# Efficacy of a Lidocaine-Impregnated Elastrator Band for Castration and Tail Docking in Lambs

**DOI:** 10.3390/ani14101403

**Published:** 2024-05-07

**Authors:** Steven M. Roche, Brenda J. Ralston, Barbara Olson, Brendan D. Sharpe, Crystal Schatz, Kendall Beaugrand, Joseph A. Ross, Madeleine A. Broomfield, Nicolas Allan, Merle Olson

**Affiliations:** 1ACER Consulting Limited, Guelph, ON N1G 5L3, Canada; 2Applied Research Team, Lakeland College, Vermilion, AB T9X 1K5, Canada; brenda.ralston@lakelandcollege.ca; 3Alberta Veterinary Laboratories, Calgary, AB T2C 5N6, Canada; barb.olson@avetlabs.com (B.O.); merle.olson@avetlabs.com (M.O.); 4Invetus Pty Ltd., Armidale, NSW 2350, Australia; bsharpe16@hotmail.com (B.D.S.); mbroomfield94@outlook.com (M.A.B.); 5Chinook Contract Research Inc., Airdrie, AB T4A 0C3, Canada; crystal.schatz@ccr01.com (C.S.); kendall@ccr01.com (K.B.); joe.ross@ccr01.com (J.A.R.); nick.allan@ccr01.com (N.A.)

**Keywords:** anesthetic, elastrator, pain control, growth, weight gain

## Abstract

**Simple Summary:**

Tail docking and castration are common procedures performed in lambs but are considered painful. Mitigation of the pain associated with these procedures is difficult, especially when using rubber ring castration. The objectives of this study were to compare castration and tail docking efficacy between lidocaine-impregnated ligation bands (LLBs) and control bands (CBs) under field conditions and identify the benefits of pain control provided by LLBs. In this study, 238 male lambs were randomly assigned to receive either LLBs or CBs on both their tail and scrotum. CBs and LLBs were both 100% effective with respect to casting success of the tail and scrotum. Lambs receiving LLBs gained more weight from d −3 to 7 following application, which may be an indication of pain control during the first week following band application. However, there were no differences observed in average daily gain over the entire study period.

**Abstract:**

The primary objective of this study was to demonstrate the non-inferiority between lidocaine-impregnated ligation bands (LLBs) and control bands (CBs) with respect to the efficacy of castration and tail docking. Secondary objectives were to compare castration and tail-docking success, evaluate local site reactions, and compare average daily gain (ADG) between the treatment groups. A total of 238 male lambs were enrolled and randomly assigned to receive LLBs or CBs on their tail and scrotum. Lambs were weighed, had a health assessment, and the band site was observed on −3, 7, 14, 21, 28, 35, and 42 days after the bands were applied. A linear regression model was built to assess average daily gain, whereas a repeated measures model was used to evaluate body weight differences at each of the measured timepoints. Furthermore, logistic regression models were used to evaluate associations with casting outcomes. Few differences were noted between treatment groups with respect to casting success for the scrotum and tail and ADG over the entire experimental period. Non-inferiority calculations demonstrated no differences in tail docking and scrotal casting success, with casting occurring for the majority of animals by d 21 and d 42 for castration and tail docking, respectively. However, lambs receiving LLBs gained more weight from d −3 to 7 (+0.03 kg/d; 95% CI: 0 to 0.07), which may be an indication of effective pain control during the first week following band application. Overall, the use of an LLB does not affect the time to successful casting of the tail and could improve short-term growth when compared to a control band. Further studies are needed to compare LLBs to multimodal methods of pain relief.

## 1. Introduction

Castration and tail docking are long-standing procedures commonly conducted in lambs. Tail docking is performed to manage fecal soiling to reduce the risk of blowfly strike and to reduce the time taken to shear sheep [1,2,3]. With regard to castration, it is practiced in ram lambs to eliminate sexual behavior and unwanted pregnancies, reduce fighting, and prevent ram taint [3]. Although alternatives are being sought to reduce and replace these procedures, they are still completed globally in millions of lambs every year.

It is established that both of these procedures are painful and stimulate nociceptors by causing tissue damage and triggering physiological pain pathways for several days [4,5,6,7,8,9,10,11]. When assessing the blood cortisol concentrations and behavioral responses of lambs following the completion of these procedures, it is common to see a range of pain responses during the first week following application [4,5,6,7,8,9,10,11]. Specifically, when lambs are tail docked and castrated, no matter the method used, active pain avoidance behaviors are observed, including kicking, rolling, tail wagging, licking, biting at the procedural site, restlessness, vocalizing, an increase in the time spent lying and in abnormal postures, and elevations in blood cortisol [4,5,6,7,8,9,10,11].

The most common technique used to complete these procedures is using an elastomeric ring, with 72% of surveyed New Zealand sheep farmers using this method for castration [8], due to the ease of application, which saves farmers time and labor costs. Although these tight rings occlude blood perfusion of the tail and scrotum, they do not prevent the conduction of nerve impulses, which can lead to a protracted pain response [9,10]. Therefore, pain control is necessary to combat not only short-term pain that results as the procedure is being conducted but also long-term pain, as it can take more than 4 weeks after the application of the elastomeric ring for castration to be complete [11].

To control the short-term pain associated with tail docking and castration, the injection of local anesthetic around the site of the band application is used. For castration, the infusion of local anesthetic into the neck of the scrotum before castration has been found to reduce the acute response associated with the procedure, including reduced cortisol and behavioral indicators of pain in the period directly following application of the band [7,10]. When local anesthetic was injected at the site of band application for tail docking or an epidural was used prior to conducting the procedure, similar responses were noted, with a reduction in cortisol and pain behavior in the hours following application [10,12]. However, the use of injectable local anesthetic has a range of associated pitfalls, including availability to farmers (lignocaine is a regulated substance in numerous jurisdictions), increased labor and time costs, and a relatively short duration of action. Pain relief in the days following the procedure can be provided using non-steroidal anti-inflammatory drugs (NSAIDs). Specifically, the use of an NSAID at the time of the procedure has been found to reduce cortisol and behavioral indicators of pain, such as restlessness, active behavior, and abnormal posture; however, the response has not been found to be consistent [12,13,14,15]. Beyond the first number of days following the completion of tail docking and castration, lambs continue to exhibit behavioral signs of discomfort prior to casting (i.e., detachment) [11]. Therefore, the development of longer-term pain control strategies is needed. Furthermore, controlling the pain resulting from surgical procedures, like castration and tail docking, can have economic benefits as a reduction in weight gain is an established outcome of pain in growing livestock [16,17,18,19,20].

Recently, a latex elastration device was developed by Chinook Contract Research Inc. that has lidocaine impregnated directly into the band [21,22,23]. This band allows for the release of lidocaine at the site of application, with the aid of a tissue permeator. Importantly, the limitations associated with the traditionally poor diffusion and delivery of lidocaine transdermally [24] have been shown to be mitigated through the use of permeability enhancers [25], which can lead to improved long-term pain relief, which has been recently demonstrated in lambs [22] and calves [23]. However, despite other studies demonstrating efficacy in offering local anesthesia starting 30 min to 28 days after application [17,22], large field studies have not been performed to evaluate the efficacy in casting success and weight gain. Hence, the primary objective of this study was to demonstrate non-inferiority between lidocaine-impregnated ligation bands (LLBs) and control bands. Secondary objectives were to compare castration and tail-docking success, evaluate local site reactions between the treatment groups, and measure the effects on average daily gain (ADG), which is a measure of pain control.

## 2. Materials and Methods

This study was conducted under University of New England Animal Ethics Committee approval (UNE AEC no. ARA22-084, approved 12OCT2022) from October to November 2022 at a commercial facility in Armidale, New South Wales, Australia.

### 2.1. Study Design

Intact male Merino lambs were enrolled at a commercial farm between 2 and 7 weeks of age. This age range was selected based on common industry practice in Australia [26], whereby sheep are farmed extensively in large groups. Ewes are typically joined for 6 weeks of age. Lambs are marked as a single group once lambing is complete; hence, 2–7 weeks covers the whole joining, allowing for the youngest lambs to be 1 week of age before mustering to the yards. Routine management practices of the commercial farm were followed, wherein study animals (ewes and lambs) grazed native and improved pastures for the duration of the study. Lambs retained ad libitum suckling access to their dam for the duration of the study. Water was provided via stock trough from farm surface water. Animals were not fasted prior to administration of the Test Articles and study activities.

### 2.2. Enrollment Criteria and Randomization

Lambs that met the enrollment criteria (2 × descended testicles, apparently healthy, weight appropriate for age (2.0 to 18.5 kg)) were enrolled 3 days prior to tail docking and castration. At enrollment, lambs were identified and weighed, and prospective scrotal and tail band sites were observed and photographed. Two hundred and forty intact male lambs split across two mobs were then randomly allocated into two groups using a randomized block design with paddock mob as a blocking factor. The control (CON) group consisted of 118 lambs, whereas the lidocaine-impregnated ligation bands (LLBs) consisted of 120 lambs. The LLBs contained 85 mg of lidocaine base USP per band (Lidoband^TM^, Solvet, AB, Canada), while the control bands were visually identical in size (13.5 mm external diameter and 6 mm internal diameter) and color and no lidocaine (Kane Veterinary Supply, Edmonton, AB, Canada). Bands were applied to the tail and scrotum for tail docking and castration, respectively, on day 0 immediately after the temperature at each site was measured with a hand-held contactless thermometer (Covetrus, Portland, OR, USA) within approximately 6–12 inches of the site. Specifically, the bands were stretched open by an applicator and slipped over the scrotum and released just above the top of the testicles (~0.5 cm). The second band was placed over the tail and released onto the tail no closer than one palpable free joint from the tail base and, ideally, three palpable free-joints from the tail base.

### 2.3. Measurements

On days 0, 7, 14, 21, 28, 35, and 42 (+/−1 day) post band-placement, animals were weighed; banding sites were observed, scored, and photographed; and temperatures at each banding site were measured (above and below each band site). Table 1 highlights the scoring for the site score and observations for the castration and tail-docking sites. For body weight measurement, scale verification was carried out by weighing standard test masses prior to and post scale use to ensure accuracy. The time to complete castration and tail docking was recorded at the weekly examinations of the lambs. Furthermore, on days 0 and 42, a complete physical exam was conducted on 25% of the animals in both treatment groups by a veterinarian.

### 2.4. Blinding

Study personnel were blinded to treatment; the control bands and lidocaine-impregnated bands were labelled as A or B, or vice versa. Both A and B bands looked identical, with no visually distinct differences that could bias or influence study participants. Personnel administrating the treatment, caring for and weighing the lambs and performing site observations (i.e., scrotal and tail inflammation scores and performing castration/tail-docking observations) were blinded to the treatment and remained blinded throughout the entire study period. Any perceived differences between bands when directly side by side were obscured once handled in isolation or when placed on the animal. Blinding was removed from all personnel once all data had been collected and statistical analysis began.

### 2.5. Sample Size Calculation

Non-inferiority was calculated for the primary endpoint (ADG) with a 15% non-inferiority margin, 90% power, and a 95% efficacy prediction in control and test bands. After accounting for a 10% drop out, a total of 90 lambs per group were needed.

### 2.6. Statistical Analysis

All statistical analyses were conducted in Stata 17 (StataCorp LP, College Station, TX, USA). Data were imported from Microsoft^®^ Excel into Stata 17 and checked for completeness. Descriptive statistics were generated on all explanatory variables in the dataset. A *t*-test was used to descriptively evaluate normally distributed data, whereas a Wilcoxon rank sum test was used to identify statistical differences between groups for non-normally distributed data. Normality was assessed using the Shapiro–Wilk test of normality. A chi-square test or Fisher’s exact test (when <5 observations in a category) was used to evaluate differences among categorical variables.

Several explanatory multivariable models were created to explore the variables contained within the dataset. Linear regression models were used to evaluate average daily gain at each of the measurement timepoints. Repeated measure linear regression models were created to evaluate the impact the treatment group had on growth over the experiment. Logistic regression models were created to evaluate the impact that the treatment group had on scrotal and tail site assessments in the experimental period when enough variation existed in the outcomes. A linear regression model was built to evaluate average daily gain over each of the time periods evaluated in this study.

Univariable regression models were constructed to screen for variables that were unconditionally associated with the outcome using a liberal *p*-value of 0.2. Risk factors that had univariate associations (*p* < 0.2) were subsequently offered to a multivariable model through a manual backward stepwise process. Evaluating the effect of the removed variables on the coefficients of the remaining variables was used to assess confounding. A variable was deemed to be a confounder if it was not an intervening variable, based on the causal diagram, and the coefficient of a significant variable in the model changed by at least 20%. Two-way interactions were evaluated between biologically important variables and remained in the final models if significant (*p* < 0.05).

For the mixed linear model, the homoscedasticity and normality of the best linear unbiased predictors (BLUPs) and residuals were evaluated for model fit. Outliers were identified and evaluated using Cook’s D, DFITS, and DFBETA. Outliers were identified and evaluated using residuals calculated for each model. If outliers were found in any of the models, they were explored to determine the characteristics of the observations that made them outliers to ensure data were not erroneous.

### 2.7. Non-Inferiority Assessment

All statistical analyses described above were conducted with the statistician blinded to the treatment. This was deliberate to eliminate any potential bias during analysis. However, to complete the non-inferiority calculations, the statistician was required to be unblinded. This only occurred when ready to complete this specific assessment.

Non-inferiority for weight gain at each time interval was measured by first identifying the 20th percentile of weight gain recorded in the control group. This was used as the benchmark for success (i.e., lambs that gained at or above the 20th percentile were considered to “pass”). The failure rate in both the control and treatment groups (i.e., the proportion of lambs that did not exceed the benchmark for success) was then determined and compared using a chi-square test. Non-inferiority for casting success of the scrotum and the tail was determined by identifying the proportion of animals in each treatment group that had tissue distal to the band at each timepoint; a logistic regression model was used to assess whether differences between treatment groups were significant.

## 3. Results

### 3.1. Descriptive Characteristics

A total of 238 lambs were enrolled in the trial, with 118 and 120 lambs randomly assigned to the CON and LLB groups, respectively. The mean (±standard deviation) weight of the lambs at d −3 to the onset of the trial was 13.2 ± 2.6 kg and was not statistically different between the treatment groups (13.3 ± 2.5 kg in CON vs. 13.2 ± 2.7 kg in LLB; *p* = 0.76). Treatments were evenly distributed across one of two paddock mobs. In the subset of 59 (29 in CON and 30 in LLB) lambs evaluated by a veterinarian on d 0 (day of band application), no differences were noted in rectal temperature (40.1 ± 0.3 °C in CON vs. 40.1 ± 0.5 °C in LLB; *p* = 0.80), heart rate (183.3 ± 32.7 beats per min in CON vs. 175.6 ± 32.3 beats per min in LLB; *p* = 0.37), and respiratory rate (80.8 ± 36.0 breaths per min in CON vs. 72.6 ± 34.8 beats per min in LLB; *p* = 0.38).

### 3.2. Growth

Average Daily Gain. Average daily gain was calculated over the following periods: d −3 to 7, d −3 to 14, d 7 to 14, d 14 to 21, d 21 to 28, d 28 to 35, d 35 to 42, and d −3 to 42. The mean growth over the periods is shown in Table 2.

D −3 to 7. Controlling for the pen of origin as a random effect, the linear regression model showed that lambs in LLB gained more weight (+ 0.03 kg/d; 95% CI: −0.0002 to 0.07; *p* = 0.049) compared to lambs in CON (Table 2; Figure 1). When evaluating for non-inferiority, the bottom 20th percentile for growth was 0.124 kg/d for lambs in CON, and 18.5% (*n* = 22) lambs in LLB were below this threshold. This was not different using a chi-square test (*p* = 0.77), indicating that the LLB is non-inferior to the CON.

D 7 to 14. In the linear regression model, controlling for the pen of origin as a random effect, no differences were noted between the treatment groups (LLB vs. CON: −0.01 kg/d; 95% CI: −0.04 to 0.02; *p* = 0.71) (Table 2). The bottom 20th percentile for growth was 0.171 kg/d for lambs in CON, and 24.6% (*n* = 29) lambs in LLB were below this threshold. This was not different using a chi-square test (*p* = 0.25), indicating that the treatment is non-inferior to the control.

D −3 to 14. No differences were found with respect to growth over this period (LLB vs. CON 0.01 kg/d; 95% CI: −0.005 to 0.03; *p* = 0.17) (Table 2; Figure 2) using a linear regression model and controlling for the pen of origin. Furthermore, the bottom 20th percentile for growth was 0.145 kg/d for lambs in CON, and 14.4% (*n* = 17) of lambs in LLB were below this threshold. This was not different using a chi-square test (*p* = 0.28), indicating that the treatment is non-inferior to the control.

D 14 to 21. Lambs in LLB gained less than those in CON (LLB vs. CON: −0.03 kg/d; 95% CI: −0.05 to −0.003; *p* = 0.02) (Table 2; Figure 3). The bottom 20th percentile for growth was 0.143 kg/d for lambs in CON, and 30.3% (*n* = 36) lambs in LLB were below this threshold. This was not significantly different using a chi-square test (*p* = 0.09), indicating that the treatment is non-inferior to the control.

D 21 to 28. In the linear regression model, controlling for the pen of origin as a random effect, no differences were found between groups (LLB vs. CON: 0.01 kg/d; 95% CI: −0.01 to 0.03; *p* = 0.55) (Table 2). The bottom 20th percentile for growth was 0.057 kg/d for lambs in CON, and 20.2% (*n* = 24) lambs in LLB were below this threshold. This was not different using a chi-square test (*p* = 0.95), indicating that the treatment is non-inferior to the control.

D 28 to 35. No differences were found between groups with respect to growth over this period (LLB vs. CON: 0.01 kg/d; 95% CI: −0.01 to 0.03; *p* = 0.36) (Table 2). The bottom 20th percentile for growth was 0.171 kg/d for lambs in CON, and 21.4% (*n* = 25) lambs in LLB were below this threshold. This was not different using a chi-square test (*p* = 0.46), indicating that the treatment is non-inferior to the control.

D 35 to 42. In the linear regression model, controlling for the pen of origin as a random effect, no differences were found between groups (LLB vs. CON: −0.002 kg/d; 95% CI: −0.03 to 0.02; *p* = 0.87) (Table 2). The bottom 20th percentile for growth was 0.086 kg/d for lambs in CON, and 23.3% (*n* = 27) lambs in LLB were below this threshold. This was not different using a chi-square test (*p* = 0.69), indicating that the treatment is non-inferior to the control.

D −3 to 42. No differences were found between groups (LLB vs. CON: 0.005 kg/d; 95% CI: −0.01 to 0.02; *p* = 0.43) (Table 2) using a linear regression model and controlling for the pen of origin as a random effect. The bottom 20th percentile for growth was 0.154 kg/d for lambs in CON, and 21.2% (*n* = 25) lambs in LLB were below this threshold. This was not different using a chi-square test (*p* = 0.80), indicating that the treatment is non-inferior to the control.

Body weight at different measurement timepoints. Body weights for each individual lamb were taken at d −3, 7, 14, 21, 28, 35, and 42 relative to enrollment. This was assessed using a repeated measures linear regression model with lamb ID as a random effect, pen and baseline weight included as fixed effects, and with the treatment by day interaction also included as a fixed effect. Treatment (*p* = 0.30) and the interaction term between treatment and day of weighing (*p* = 0.57) were not significant; however, the day of weighing was significant (*p* < 0.0001).

### 3.3. Scrotal Site Assessment

Erythema, swelling, site temperature score, and target appearance and feel. At d 7, no differences were noted with respect to the presence of erythema (1.7% in CON vs. 0.8% in LLB; *p* = 0.62) and swelling (0% in CON vs. 0.8% in LLB; *p* = 1.00) using a Fisher’s exact test. Beyond d 7, none of the enrolled lambs had erythema or swelling. Furthermore, none of the lambs had an elevated site temperature score, whereas all lambs that had tissue distal to the target area presented with tissue that was shriveled and/or cold at each of the timepoints evaluated.

Tissue temperature above the band. Table 3 below highlights the tissue temperature that was above the band application site. On d 14, it was found that lambs in LLB had a lower temperature above the band application site; however, no other differences in temperature at the other timepoints were noted.

Presence of the band with tissue distal to castration site. The proportion of lambs in the different treatment groups with the presence of the band with tissue distal to the site of castration is presented in Table 4. Differences were noted between groups with regard to timepoints in terms of casting success. Specifically, at d 21, the presence of the band with tissue distal to the target area tended to be different between groups (LLB vs. CON: Odds Ratio (OR): 1.82; 95% CI: 0.92 to 3.60; *p* = 0.09) using a logistic regression model with the pen as a fixed effect. While a statistical tendency was found, using a *p*-value of 0.05, these results would suggest the treatment is non-inferior to the control. Furthermore, at d 28, lambs in LLB had 2.06-times (95% CI: 1.22 to 3.47; *p* = 0.007) greater odds of having tissue distal to the band compared to CON lambs at d 28 using a logistic regression model with the pen as a fixed effect, suggesting that this treatment is inferior at this timepoint when compared to the control at this timepoint. At d 35, lambs in LLB tended to have higher odds of having tissue distal to the band (OR: 1.95; 95% CI: 0.99 to 3.85; *p* = 0.05) compared to CON; however, at d 42, no differences were noted between groups (*p* = 0.37).

### 3.4. Tail Site Assessment

As all animals had successful casting of the tail by d 28, descriptive statistics are not presented from d 28 onwards for tail site assessment.

Erythema, swelling, site temperature score, and target appearance and feel. At d 7, no differences were noted between groups with respect to the presence of erythema (7.0% in CON vs. 10.1% in LLB; *p* = 0.49) and swelling (7.0% in CON vs. 4.2% in LLB; *p* = 0.36) using a chi-square or Fisher’s exact test. Beyond d 7, none of the enrolled lambs had erythema or swelling. Furthermore, none of the lambs had an elevated site temperature score, whereas all lambs that had tissue distal to the target area showed tissue that was shriveled and/or cold at each of the timepoints evaluated.

Tissue temperature above the band. No differences were noted with regard to the tissue temperature of the tail above the band at any of the measured timepoints (Table 5).

Presence of the band with tissue distal to tail docking site. The proportion of lambs in the different treatment groups with the presence of the band with tissue distal to the site of castration is presented in Table 6. No statistical differences were noted at any of the measured timepoints.

### 3.5. Veterinary Exam at d 42

In the subset of 57 (27 in CON and 30 in LLB) lambs evaluated by a veterinarian on d 42, no differences were noted in rectal temperature (39.6 ± 0.4 °C in CON vs. 39.7 ± 0.4 °C in LLB; *p* = 0.28), heart rate (138.2 ± 18.9 beats per min in CON vs. 140.8 ± 29.6 beats per min in LLB; *p* = 0.70), and respiratory rate (92.3 ± 27.2 breaths per min in CON vs. 94.4 ± 25.2 beats per min in LLB; *p* = 0.76).

## 4. Discussion

Previous work on the LLB used in the present study has demonstrated that these bands are able to effectively achieve therapeutic doses in the tissues studied over a 28-day period [21,22,23]. The goal of the present study was to compare castration and tail-docking efficacy between lidocaine-impregnated ligation bands and control bands under field conditions and to identify the benefits of pain control provided the LLB, with a focus on time to casting and ADG.

In this study, the LLB was found to be non-inferior to the control bands in nearly all parameters evaluated. However, differences were noted in the first week following band application, where lambs in the LLB gained more body weight over that period, which is considered the most painful period following castration and tail docking [4,5,6,7,8,9,10,11,12]. It is well known that a reduction in ADG is an indication of pain [16,17,18,19,20], and a positive improvement in ADG could reflect improved pain management. It was shown, in a study by Ross et al. [22], that tissue lidocaine concentrations started to reach the effective concentration for pain control (EC50 or EC95) at 30 min after band placement for tail and scrotal tissues, suggesting immediate pain control after band application. Control bands showed an increase in ADG between d 14 to 21 following castration and tail docking compared to LLB-treated animals. Other studies have noted that, in the absence of analgesia, a reduction in feeding behavior and growth occurs over the subsequent days following the procedure; however, compensatory growth in the weeks following the procedure occurs, as most of the pain has diminished from the tail and scrotum site [18,23]. Although compensatory weight gain occurs in control lambs, it is possible that carcass quality (fat vs. lean muscle and muscle development) may be influenced [27]. Future studies should evaluate the effects of LLBs on a larger group of animals, where body weight gain, carcass quality, and feed efficiency are measured to further explore these findings.

It has been noted that local anesthetics have varying effects on wound healing, with some noting that they inhibit collagen and glycosaminoglycan synthesis, which are crucial for the healing process [28,29]. Additionally, lidocaine has been found to affect collagenization at the wound site [30]; however, it is highly dependent on the concentrations of lidocaine, with lower concentrations found not to impair wound health [31]. Although the process of castration with a band involves chronic ischemia and tissue necrosis, it is possible that the lidocaine in the LLB may delay casting; however, it is important to note that no differences were found at any timepoint with regard to casting the tail or at d 42 with respect to casting the scrotum. Furthermore, in other studies that used topical anesthesia at the site of castration, no differences were noted in the time to successful castration [16]. Therefore, further studies should evaluate whether the delay in casting is clinically relevant and the physiological processes involved in the delay.

There are some limitations to consider when interpreting the results of this study. First, this study detected the non-inferiority between LLBs and a control band regarding growth. Hence, interpretations of the results of this study should be made with this consideration. No behavioral or physiological measures of pain were evaluated in this study, which would have provided insight into the utility of the LLBs in controlling the pain associated with tail docking and castration. As this was a field study to compare castration and tail-docking efficacy and ADG between LLB and CON bands under commercial conditions, typical physiological and behavioral measurements could not be performed. There was also no positive control group used in this study, where multimodal pain relief, such as the use of a local anesthetic block and NSAID, was provided as a treatment group. Future studies should consider comparing the use of an LLB with multimodal pain relief to compare the growth performance and behavioral and physiological measures of pain between the groups. Future studies should also consider the potential uses of other forms of longer-acting local anesthetic agents, such as bupivacaine, which has been demonstrated to have a longer duration than lidocaine [32,33,34,35]. For example, Melches et al. [36] observed fewer abnormal behaviors in lambs undergoing ring castration following the administration of bupivacaine compared to lambs receiving lidocaine. Therefore, additional research that considers comparisons between LLBs and positive controls receiving longer-acting local anesthetics to evaluate their efficacy against other reasonable pain mitigation options may be warranted.

Despite the inconsistent trend in superiority observed between the LLB and CON groups (though non-inferiority was the primary objective), our study suggests that products such as the LLB evaluated here offer a means for farmers to effectively mitigate the pain associated with castration and/or tail docking while minimizing the labor and time costs associated with offering pain control when using conventional bands, though it is important to note that more research is needed to corroborate and further elucidate the effects described in this study.

## 5. Conclusions

This study found that the LLBs were non-inferior to control elastomeric bands in the time to casting the tail and growth over the entire experimental period. There were differences noted in growth during the most painful period following application, where lambs that were castrated and tail docked with the LLBs had greater growth in the week after the procedure but lower growth between weeks 2 and 3 of the trial. Furthermore, casting of the scrotum was found to be different, taking slightly longer to occur in lambs that had LLBs, which may be an indication of reduced inflammation or irritation with the LLB. Future research should compare the use of this novel LLB to multimodal pain relief to evaluate the efficacy of each method in controlling the pain associated with castration and tail docking.

## 6. Patents

US11596510, CA3072762, AU2018313951, NZ2762352, AU202214318, NZ430877, EP3664720A4, and WO2019032928A1.

## Figures and Tables

**Figure 1 animals-14-01403-f001:**
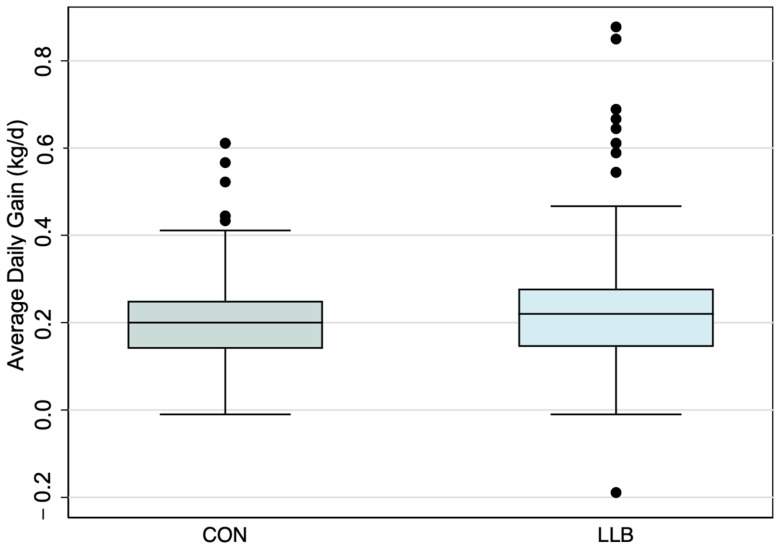
Boxplot highlighting the average daily gain from d −3 to 7 after enrollment, by treatment group.

**Figure 2 animals-14-01403-f002:**
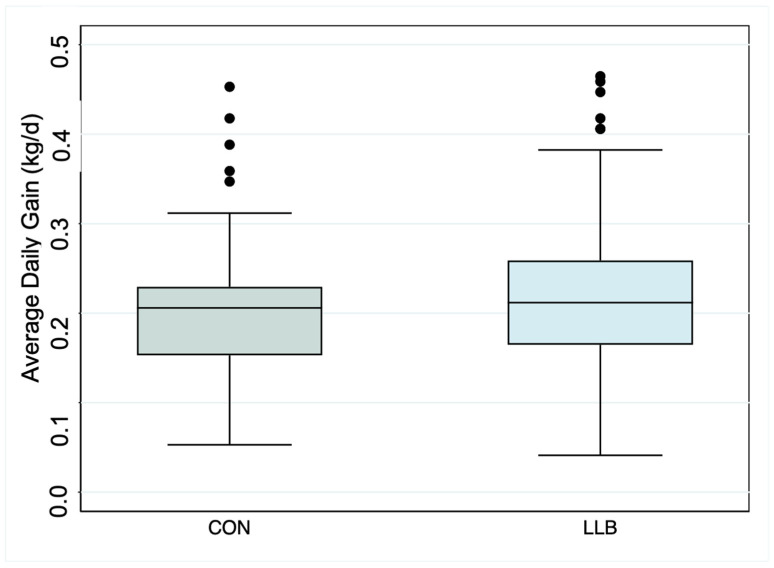
Boxplot highlighting the average daily gain from d −3 to 14 after enrollment, by treatment group.

**Figure 3 animals-14-01403-f003:**
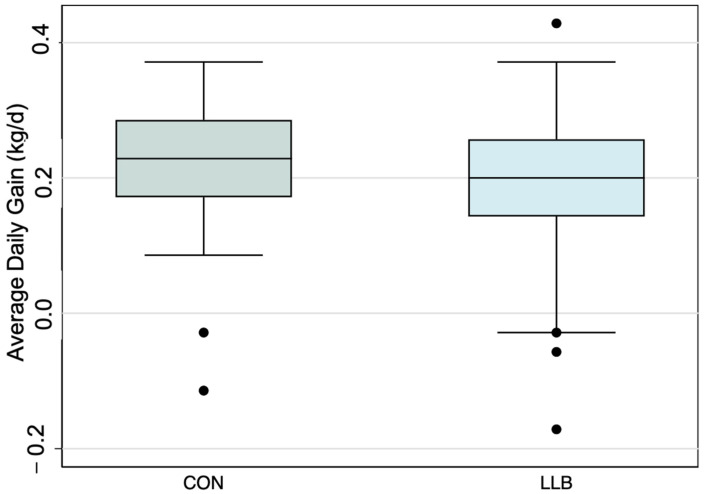
Boxplot highlighting the average daily gain from d 14 to 21 after enrollment, by treatment group.

**Table 1 animals-14-01403-t001:** Scoring for site reaction scores and site observations for castration and tail-docking site.

	Local Site Reaction Score		Castration and Tail-Docking Site Observation	
Score	Erythema and Swelling	Site Temperature Palpation	Target Area Appearance/Feel	Band Target Site Observations
0	Absent	Normal	Tissue distal to the target area has shriveled and/or is cold (indicating that the tissue is non-viable)	Band is absent with absence of tissue distal to the target area (normal castration or tail docking outcome) OR band is present with presence of unviable tissue distal to the band and casting is achieved via gentle manipulation (i.e., casting was not achieved due to overlaying fleece contact)
1	Mild	Elevated	Tissue distal to the target area is normal in appearance and normal body temperature.	Band is present with presence of tissue distal to the target area (castration or tail docking not yet complete)
2	Moderate			Only the band is present, the tissue distal to the target area is absent (scrotum/tail has been cast but the band is still at the site)
3	Severe			Band is absent with presence of the non-viable tissue distal to the target area (band has prematurely broken but scrotal tissue or tail tissue is non-viable)
4	Severe with exudate			Band is absent with presence of the viable tissue distal to the target area (band has prematurely broken if post-band placement observation)
Composite Score	Sum of the score, per site, for Erythema, Swelling and Heat.		Sum of the score, per site, for Target Area Appearance/Feel and Band Target Site Observation.	

**Table 2 animals-14-01403-t002:** Average daily gain (ADG) (kg/d) of LLB and CON lambs over the measured timepoints in the experiment. The mean ± standard deviation is presented for each timepoint by treatment group along with the corresponding *p*-value from the linear regression model, controlling for pen of origin as a random effect.

Time Period	LLB	CON	*p*
d −3 to 7	0.24 ± 0.17	0.21 ± 0.11	0.049
d 7 to 14	0.21 ± 0.14	0.22 ± 0.08	0.71
d −3 to 14	0.22 ± 0.09	0.20 ± 0.07	0.17
d 14 to 21	0.20 ± 0.09	0.22 ± 0.08	0.02
d 21 to 28	0.13 ± 0.08	0.13 ± 0.09	0.55
d 28 to 35	0.26 ± 0.11	0.25 ± 0.08	0.36
d 35 to 42	0.17 ± 0.11	0.18 ± 0.08	0.87
d −3 to 42	0.20 ± 0.07	0.20 ± 0.04	0.43

**Table 3 animals-14-01403-t003:** Tissue temperature above the band application site on the scrotum at each of the timepoints evaluated. The mean ± standard deviation is presented for normally distributed variables, whereas the median (range) is presented for non-normally distributed variables. The *p*-values were generated using a *t*-test and Wilcoxon rank sum test for normal and non-normal variables, respectively.

Time Period	LLB	CON	*p*
d 7	37.7 °C (35.1 to 39.9 °C)	37.7 °C (34.5 to 39.8 °C)	0.76
d 14	37.1 °C (34.3 to 40.6 °C)	37.7 °C (34.6 to 41.3 °C)	0.04
d 21	37.5 °C (34.1 to 40.2 °C)	37.2 °C (34.0 to 39.4 °C)	0.38
d 28	37.7 ± 1.1 °C	37.7 ± 1.1 °C	0.92
d 35	37.2 °C (34.2 to 39.1 °C)	37.1 °C (34.1 to 39.7 °C)	0.88
d 42	36.7 °C (34.3 to 40.0 °C)	36.7 °C (34.0 to 39.6 °C)	0.53

**Table 4 animals-14-01403-t004:** The number of lambs observed and % (*n*) of lambs with presence of the band with tissue distal to castration site at each of the timepoints evaluated. The *p*-values were generated, at time-points with enough variability, using logistic regression model with pen as a fixed effect.

Time Period		LLB		CON		*p*
		*n*	% (*n*) with Presence of Band and Tissue	*n*	% (*n*) with Presence of Band and Tissue	
d 7		119	100% (119)	115	100% (115)	-
d 14		116	99.1% (115)	113	96.5% (109)	0.21
d 21		119	86.6% (103)	118	78.0% (92)	0.09
d 28		119	53.8% (64)	116	36.2% (42)	0.007
d 35		117	23.9% (28)	115	13.9% (16)	0.05
d 42		118	3.4% (4)	116	0.9% (1)	0.37

**Table 5 animals-14-01403-t005:** Tissue temperature above the band application site on the tail at each of the timepoints evaluated. The median (range) is presented, and the *p*-values were generated using a Wilcoxon rank sum test.

Time Period	LLB	CON	*p*
d 7	38.5 °C (35.3 to 40.8 °C)	38.7 °C (34.1 to 40.6 °C)	0.64
d 14	37.2 °C (34.1 to 41.0 °C)	37.4 °C (34.2 to 40.1 °C)	0.71
d 21	37.0 °C (34.1 to 40.5 °C)	37.0 °C (34.0 to 40.7 °C)	0.85

**Table 6 animals-14-01403-t006:** The number of lambs observed and % (*n*) of lambs with presence of the band with tissue distal to tail docking site at each of the timepoints evaluated. The *p*-values were generated, at time-points with enough variability, using logistic regression model with pen as a fixed effect.

Time Period		LLB		CON		*p*
		*n*	% (*n*) with Presence of Band and Tissue	*n*	% (*n*) with Presence of Band and Tissue	
d 7		119	100% (119)	115	100% (115)	-
d 14		116	72.4% (84)	113	63.7% (72)	0.21
d 21		119	0.8% (1)	118	0% (0)	-

## Data Availability

Data will be made available upon request.

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
