# Peer review of "Efficacy of a Lidocaine-Impregnated Elastrator Band for Castration and Tail Docking in Lambs"

_animals, 2024, doi:10.3390/ani14101403_

Round 1

Reviewer 1 Report

Comments and Suggestions for Authors

Dear Authors,

This manuscript is well-written and it give potential promising insights. 

However, i think some corrections are necessary.

- Line 37. "following band application" is a repetition

- Line 37. Indicate in the abstract how many days are necessary to casting.

- Line 47. [1-3] instead of [1,2,3]

- Line 68 and following. As lidocaine normally does not have a good trancutaneous absorption, a description and a reference about this should be added.

- Line 90. [22-24] instead of [22,23,24]

- Line 89 and following. A deeper explanation about the method of slow release of this device should be added. 

- Line 96. Is CB repeated in the text? If not, it should be deleted.

- Line 119 and following. The control band's manifacturer name is not indicated. Also, its characteristics are not indicated.

- Line 124. Indicate the manufacturer's name of the thermometer. Also, indicate the distance of measuring.

- Table 1. Why are many slots blank?

- Line 142 and following. The method for blinding is not so clear with this paragraph. It should be stated more clearly.

- Line 301. Is "noted" meant instead of "not"?

- Lines 381-382. A repetition of "that" is present.

- Line 390. How would the carcass be influenced? This information should be added.

- As the LLBs are non-inferior, I think that a discussion about the pros and cons of their use should be added. For example, a discussion about the costs and applicability should be made.

Best regards

Author Response

Reviewer 1:

This manuscript is well-written and it give potential promising insights. However, i think some corrections are necessary.

AU: Thank you for your thoughtful comments. We have addressed all notes below and feel the manuscript has improved as a result. We appreciate your time and effort. 

R: Line 37. "following band application" is a repetition

AU: We have revised.

R: 37. Indicate in the abstract how many days are necessary to casting.

AU: We have revised (Lines 34-36)

R: Line 47. [1-3] instead of [1,2,3]

AU: Corrected

R: Line 68 and following. As lidocaine normally does not have a good trancutaneous absorption, a description and a reference about this should be added.

AU: We have added references to this issue, the effect permeability enhancers can have to overcome this issue, and to our previous work demonstrating effective absorption using this method (Lines 91-96)

R: Line 90. [22-24] instead of [22,23,24]

AU: Corrected

R: Line 89 and following. A deeper explanation about the method of slow release of this device should be added.

AU: We have added some references and additional detail that better describe the band and its properties (Lines 91-96)

R: Line 96. Is CB repeated in the text? If not, it should be deleted.

AU: Corrected

R: Line 119 and following. The control band's manifacturer name is not indicated. Also, its characteristics are not indicated.

AU: We have provided additional details in Lines 125-128

R: Line 124. Indicate the manufacturer's name of the thermometer. Also, indicate the distance of measuring.

AU: We have added details as requested (Lines 130-131)

R: Table 1. Why are many slots blank?

AU: The cells that appear to be blank are actually merged together because the content contained within the merged cell pertains to all rows within that section. We believe this is just a formatting aspect that can be addressed by the editors when developing the final proof.

R: Line 142 and following. The method for blinding is not so clear with this paragraph. It should be stated more clearly.

AU: We have modified to help make it clearer (Lines 150-158)

R: Line 301. Is "noted" meant instead of "not"?

AU: Yes, this has been corrected.

R: Lines 381-382. A repetition of "that" is present.

AU: Corrected.

R: Line 390. How would the carcass be influenced? This information should be added.

AU: We have elaborated as suggested (Lines 393-395)

R: As the LLBs are non-inferior, I think that a discussion about the pros and cons of their use should be added. For example, a discussion about the costs and applicability should be made.

AU: We have provided some additional commentary on this in the discussion (Lines 423-429)

Reviewer 2 Report

Comments and Suggestions for Authors

This is an interesting study on a  control  investigation using rubber rings for detailing and castration of lambs .The use of rubber rings impregnated with lidocaine was compared to a control.

There are a number of points which require clarification.

1 The age of the lambs was stated as 2 to 7 weeks .Some countries specify that the  procedure should not be carried out after 7 days of age .Can the authors justify their choice of ages

2.Is there any information as to how long the lidocaine persists in the rings once applied to the lambs'tissues.Was any blood concentration of lidocaine measured to assess any systematic effects.

3.Would a cintrol study involving using the lidocaine ring to the tail and the control to thev scrotum and vice versa.

4.Has consideration been given to the possible use of a longer acting local anaesthetic agent?

Lines 53 and 58  -check the correct spelling of behaviour and behavioural

Comments on the Quality of English Language

No attention required

Author Response

This is an interesting study on a control investigation using rubber rings for detailing and castration of lambs. The use of rubber rings impregnated with lidocaine was compared to a control.

AU: Thank you for your thoughtful comments, we appreciate you taking the time to review our manuscript. We have offered some revisions and responses below and within the manuscript that we hope you feel will satisfactorily address your thoughts. 

There are a number of points which require clarification.

R: 1 The age of the lambs was stated as 2 to 7 weeks. Some countries specify that the procedure should not be carried out after 7 days of age. Can the authors justify their choice of ages

AU: Thank you for this comment. 2-7 weeks of age is selected based industry practice in Australia, whereby sheep are farmed extensively in large groups. Ewes are typically joined for 6 weeks of age. Lambs are marked as a single group once lambing is complete, hence 2-7 weeks covers the whole joining, allowing for the youngest lambs to be 1 week of age before mustering to the yards. Multiple musters during the lambing period is avoided due to significant risk of mismothering in Merino sheep

R: Is there any information as to how long the lidocaine persists in the rings once applied to the lambs'tissues. Was any blood concentration of lidocaine measured to assess any systematic effects.

AU: The work cited throughout our study (22-24 in particular) describes past work on this product that details the correlation between lidocaine in the band and in the tissue itself. No published work is available on blood concentration, though systemic effects of this product are currently being investigated in an ongoing study.

R: Would a cintrol study involving using the lidocaine ring to the tail and the control to thev scrotum and vice versa.

AU: There could be merit in this type of study, particularly if the goal were to measure tissue levels locally or systemically. However, this was not a stated objective of our work. Further, this study design would require a doubling of our sample size, which we believe would be challenged by our animal ethics committees regarding the need and validity of causing more discomfort to a larger population of animals.

R: Has consideration been given to the possible use of a longer acting local anaesthetic agent?

AU: This has not been a consideration at this time, largely because most other products use lidocaine at this time and this study was focused on the agent used within the commercially available product. Additionally, the papers we have cited (22-24) that evaluated this product in lambs and calves demonstrated that accumulates over 28 days.

R: Lines 53 and 58 -check the correct spelling of behaviour and behavioural

AU: Corrected

Round 2

Reviewer 2 Report

Comments and Suggestions for Authors

See previous assessment

Author Response

AU: We appreciate the reviewer’s feedback and have provided our explanations for each point raised below. We have modified the manuscript in several areas to reflect areas that could benefit from more clarification and discussion, and as a result, feel the manuscript is significantly improved.

Reviewer 2:

This is an interesting study on a control investigation using rubber rings for detailing and castration of lambs. The use of rubber rings impregnated with lidocaine was compared to a control.

There are a number of points which require clarification.

R: 1 The age of the lambs was stated as 2 to 7 weeks. Some countries specify that the procedure should not be carried out after 7 days of age. Can the authors justify their choice of ages

AU: Thank you for this comment. 2-7 weeks of age is selected based industry practice in Australia, whereby sheep are farmed extensively in large groups. Ewes are typically joined for 6 weeks of age. Lambs are marked as a single group once lambing is complete, hence 2-7 weeks covers the whole joining, allowing for the youngest lambs to be 1 week of age before mustering to the yards. Multiple musters during the lambing period is avoided due to significant risk of mismothering in Merino sheep

Further, we have modified the methods section to provide the reader with more description of this justification. We have also referenced the Meat and Livestock Australia’s report on best husbandry practices. Lines 111-115.

R: Is there any information as to how long the lidocaine persists in the rings once applied to the lambs'tissues. Was any blood concentration of lidocaine measured to assess any systematic effects.

AU: The work cited throughout our study (22-24 in particular) describes past work on this product that details the correlation between lidocaine in the band and in the tissue itself. No published work is available on blood concentration, though systemic effects of this product are currently being investigated in an ongoing study. We have added an explanation of these effects and pointed to this past work in the discussion (Lines 384-388)

R: Would a cintrol study involving using the lidocaine ring to the tail and the control to thev scrotum and vice versa.

AU: We appreciate this suggestion. We believe there could be merit in this type of study, particularly if the goal were to measure tissue levels locally or systemically. However, this was not a stated objective of our work for this particular manuscript; though it is the subject of ongoing work. Further, this study design would require a doubling of our sample size, which we believe would be challenged by our animal ethics committees regarding the need and validity of causing more discomfort to a larger population of animals. As a result, we feel the approach used here is justifiable and we are pleased to report that a future study will report on the use of this methodology in the pursuit of a different outcome related to LLBs.

R: Has consideration been given to the possible use of a longer acting local anaesthetic agent?

AU: Our initial interpretation of this comment was incorrect, we apologize for the misunderstanding. We had initially suggested that the development of an LLB using a longer-acting local anesthetic had not been considered, largely because most other products use lidocaine. We had also noted that the focus of this paper was to investigate the effect of the commercially available product itself. However, we appreciate the comment may be more focused on the potential to compare the LLB to a longer-acting local, opposed to a negative control. We do feel this would be worthwhile for future studies and have now discussed this in Lines 432-439.

R: Lines 53 and 58 -check the correct spelling of behaviour and behavioural

AU: Corrected